# Comparison between 20 and 30 meters in walkway length affecting the 6-minute walk test in patients with chronic obstructive pulmonary disease: A randomized crossover study

**Narongkorn Saiphoklang**[1,2]*, **Apiwat Pugongchai**[2], **Kanyada Leelasittikul**[2]

1 Division of Pulmonary and Critical Care Medicine, Department of Internal Medicine, Faculty of Medicine, Thammasat University, Pathum Thani, Thailand, 2 Medical Diagnostics Unit, Thammasat University Hospital, Pathum Thani, Thailand

* m_narongkorn@hotmail.com

## Abstract

### Background

A 30-m walkway length for the 6-minute walk test (6MWT) is the standard recommendation established by the American Thoracic Society to assess patients with chronic obstructive pulmonary disease (COPD). This study aimed to compare between the distances of 20 and 30 m long corridor affecting 6MWT in COPD patients.

### Methods

A randomized crossover study was conducted with patients. COPD patients were randomized 1:1 to either a 20-m or a 30-m walkway in the first test, then switched to the other in the second test. Physiologic parameters and 6-minute walking distance (6MWD) were recorded.

### Results

Fifty subjects (92% men) were included: age 69.1±7.4 years, body mass index 22.9±5.5 kg/m$^2$, $FEV_1$ 63.0±21.3%, and 50% having cardiovascular disease. The 6MWD in a 20-m and a 30-m walkway were 337.82±71.80 m and 359.85±77.25 m, respectively (P<0.001). Mean distance difference was 22.03 m (95% CI -28.29 to -15.76, P<0.001). Patients with a 20-m walkway had more turns than those with a 30-m walkway (mean difference of 4.88 turns, 95% CI 4.48 to 5.28, P<0.001). Also, higher systolic blood pressure was found in patients with a 20-m walkway after 6MWT (4.62 mmHg, P = 0.019). Other parameters and Borg dyspnea scale did not differ.

**Funding:** The work was supported by Thammasat University Hospital, Thailand. The funders had no role in study design, data collection and analysis, decision to publish, or preparation of the manuscript.

**Competing interests:** The authors have declared that no competing interests exist.

## Conclusions

The walkway length had significant effect on walking distance in COPD patients. A 30-m walkway length should still be recommended in 6MWT for COPD assessment.

## Clinical trial registration

**Clinicaltrials.in.th number:** TCTR20200206003.

## Introduction

Chronic obstructive pulmonary disease (COPD) is a progressive, irreversible inflammatory disease that makes it difficult to breathe. It was the fourth leading cause of death in the world by 2020 [1]. Common symptoms include a chronic cough, dyspnea, more sputum secretion, lung wheezing, and chest tightness. It's caused by exposure to tobacco smoke, occupational chemicals, indoor and outdoor air pollution [1], and genetics [1, 2]. In patients with disease progression COPD limits physical activity and is related to a higher risk of exacerbation and increased risk of mortality [3, 4]. Therefore, assessment of physical activity is important for clinical management of COPD patients.

The 6-minute walk test (6MWT) is submaximal exercise that corresponds to functional activity used in daily activities. 6MWT is an effective exercise which is low cost, uncomplicated and provides a measure for evaluation of cardiopulmonary, musculoskeletal and nervous systems. In clinical practice, 6MWT is often used to evaluate patients with various diseases, especially COPD. 6MWT is a tool for evaluating physical activities, comparing pre-post treatment and predicting morbidity and mortality rates in COPD patients [5]. A guideline of the American Thoracic Society (ATS) suggests the walkway distance should not be less than 30-m [5]. Previous studies reported that a difference of 10 meters in the total walking distance affects the 6MWT results [6, 7]. In contrast, another study found that course length had no significant effect on walking distance 50 feet (15.24 m) to 164 feet (49.99 m) [8]. However, there are conflicting data on different walkway lengths of 6MWT in the COPD patient population with varying ethnicities. The aim of this study was to measure the difference of walking distance and other effects in walkways of 20-m and 30-m.

## Methods

### Study design and participants

A randomized crossover study of COPD subjects was conducted in Thammasat University Hospital, Thailand between June 2018 –January 2019. COPD diagnoses using post-bronchodilator forced expiratory volume in 1 second to forced vital capacity ($FEV_1/FVC$) < 0.7 were reviewed by chest physicians. The inclusion criteria were Thai COPD patients, age 40 to 80 years, smoking history ≥10 pack-years, and post-bronchodilator $FEV_1/FVC$ < 0.7. The exclusion criteria were recent COPD exacerbation, eye surgery or abdominal surgery within 6 weeks, inability to walk or failure to cooperate in walking, $O_2$ saturation < 90% in resting stage, blood pressure below 90/60 or above 180/100 mmHg prior to testing, heart rate less than 50 or more than 120 beats per minute, disease other than COPD; e.g. lung cancer, pulmonary fibrosis, and pulmonary tuberculosis, myocardial infarction. Patients' characteristics, treatment profiles, exacerbation history and comorbidities were collected.

This study was approved by the Institutional Review Board of the Human Ethics Committee of Thammasat University No.2 (Project No. 001/2561, Certificate of Approval CoA 005/2561) (see S1–S6 Files). This study was registered in the Thai Clinical Trials Registry (TCTR); thaiclinicaltrials.org (number: TCTR20200206003). All participants provided written informed consent. This study started the first enrollment on June 1, 2018 and completed the follow-up on March 31, 2019. However, this study was not registered before enrolment of participants started because we initially judged that the study was an observational study rather than an interventional trial. The authors confirm that all ongoing and related trials for this intervention are registered.

## Procedures and outcomes

Each patient performed the 6MWT twice in accordance with the ATS 2002 guidelines [5]. Patients were randomized 1:1 with block of four using a computer program to a 20-m walkway or a 30-m walkway in the first test, then the patient rested until all vital signs had stabilized. The other walkway test was performed in the second test (see Fig 1). In each test, bronchodilator was administered before each walk and the patient had to walk as fast as possible in a 6-minute period. The patient was encouraged walk the entire distance (20 or 30m). The patient was allowed a period of rest if shortness of breath developed. We collected demographics, spirometry variables, and physiologic variables including blood pressure, heart rate, respiratory rate, oxygen saturation and 10-point Borg dyspnea scale. After each test, we measured the same data variables as before test and recorded the number of turns, and 6-minute walking distance (6MWD). The patients were allowed to stop or rest during testing. All 6MWDs were recorded.

## Statistical analysis

Based on a previous study [9], the difference in 6MWD between 10-m walkway and a 30-m walkway in COPD patients was 49.5±33.6 meters. We hypothesized that the difference in 6MWD between 20-m walkway and a 30-m walkway in those patients was 34 meters. Thus, 50 patients were needed to be studied with 90% power and 5% type I error.

Statistical analyses were performed using SPSS version 23.0 software (IBM Corp., Armonk, NY, USA). Data is presented as mean ± standard deviation and number (%). Paired t-test was used to compare continuous variables between two groups. A two-sided p-value < 0.05 was considered statistically significant.

## Results

Fifty-nine patients were screened. Of these, 50 patients were eligible for inclusion (see Fig 1). Males numbered 46 (92%). Mean age was 69.1±7.4 years. $FEV_1$ was 63.0±21.3%. The patients having cardiovascular disease, dyslipidemia and lung disease other than COPD were 50%, 38% and 16%, respectively. The characteristics of the patients are shown (see Table 1).

We found that the 6MWD in a 20-m and a 30-m walkway was 337.82±71.80 m and 359.85±77.25 m, respectively (P<0.001). Mean distance difference was 22.03 m (95% CI -28.29 to -15.76, P<0.001). Patients had 16.46±3.51 turns in 20-m walkway and 11.58±2.56 turns in 30-m walkway (mean difference of 4.88 turns, 95% CI 4.48 to 5.28, P<0.001).

Subsequently, we studied differences of effects of walkway distance on multivariable data include Borg dyspnea scale and physiologic variables. Systolic blood pressure was found to be higher in patients with a 20-m walkway after 6MWT (4.62 mmHg, 95% CI 0.77 to 8.46, P = 0.019). Other physiologic variables are shown in Table 2. All physiologic variables of perceived exertion after testing were higher than before testing in the 20-m test, but the 30 m test

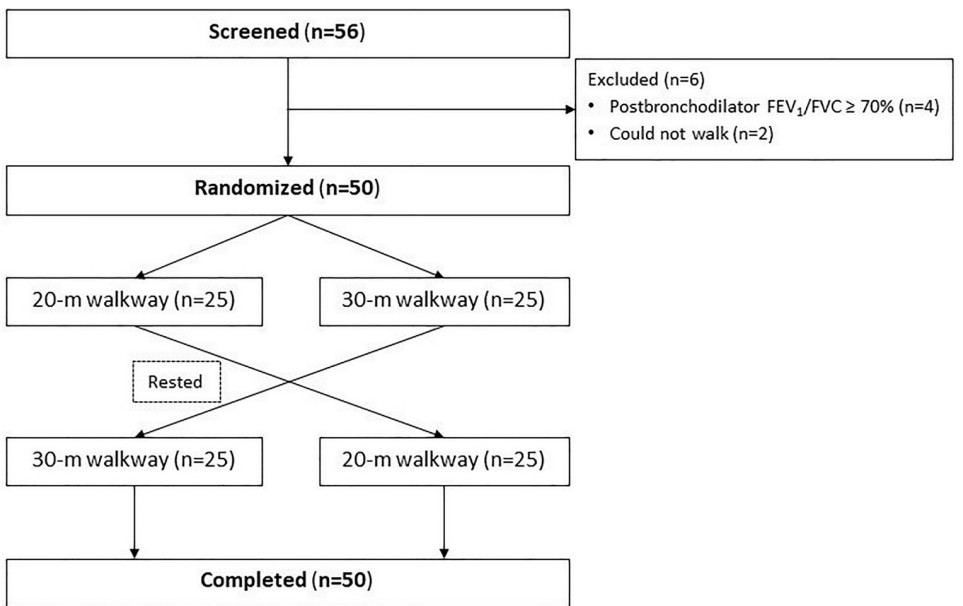

**Fig 1. Flow chart of participant recruitment to the study.**

did not seem to affect diastolic blood pressure (see Table 3). The dataset of 6MWT in COPD patients is shown in S7 File.

## Discussion

This is the first randomized crossover study to compare between the distances of 20-m and 30-m long corridors affecting 6MWT in COPD patients. Our study found significant difference of 6MWD between a 20-m and a 30-m walkway in that walking distance on 20-m walkway is shorter than on a 30-m by an average of 22.03 m and had 4.88 more turns. These results correspond to the study of Klein SR, et al [10]. They found that, in COPD patients, 6MWD with 30-m walkway was significantly longer than 20-m walkway (mean difference in 22.1 m) [10]. We think that our sample size has sufficient power to detect a statistically significant difference by the 2-sided test for walking distance between two walkway lengths. Sample size estimation was based on findings from a previous study of Beekman E, et al [9].

ATS 2002 guidelines recommended at least 30-m corridors for 6MWT [5]. The study of Ng SS and coworkers, studied 25 healthy subjects aged 50 years and over to compare the walkway lengths of 10-m, 20-m and 30-m. Their results revealed that the walkway length affects the total walking distance [7], which is consistent with the study of Beekman E and coworkers comparing 10-m and 30-m corridors for COPD patients. They found that the average walking distance on the 10-m was less than on the 30-m by (approximately 49.50-m) [9]. These findings were similar to our study because the longer walkways have fewer turns than shorter walkways. In contrast, the study of Sciurba F and coworkers showed that the effect on walking distance of walkway lengths of 50 feet (15.24-m) to 164 feet (49.99-m) made no difference [8]. There are also studies of patients with chronic stroke and cirrhotic patients waiting for liver transplants. In chronic stroke patients, the study by Ng SS and coworkers showed significant difference between walkway lengths of 10-m, 20-m and 30-m [6], but the study of cirrhotic patients by Veloso-Guedes CA and coworkers found no difference of effect between 20-m and 30-m [11].

**Table 1. Baseline characteristics of the COPD patients.**

| Characteristics | N = 50 |
|---|---|
| Age, years | 69.1 ± 7.4 |
| Male/Female | 46/4 (92.0/8.0) |
| Body mass index, kg/m$^2$ | 22.9 ± 5.5 |
| Smoking history, pack-years | 32.7 ± 23.0 |
| Active smoking | 15 (30) |
| $FEV_1/FVC$, % | 59.8 ± 10.8 |
| $FEV_1$, L | 1.50 ± 0.55 |
| $FEV_1$, % predicted | 63.05 ± 21.29 |
| FVC, L | 2.45 ± 0.74 |
| FVC, % predicted | 78.20 ± 19.43 |
| Patients with exacerbation history | 7 (14.0) |
| **COPD medications** | |
| LABA | 2 (4.0) |
| LAMA | 19 (38.0) |
| LABA and LAMA | 6 (12.0) |
| LABA and ICS | 27 (54.0) |
| SABA | 1 (2.0) |
| SABA and SAMA | 30 (60.0) |
| Xanthine | 19 (30.0) |
| Oral prednisone | 1 (2.0) |
| Oral N-Acetylcysteine | 23 (46.0) |
| Long-term Azithromycin | 3 (6.0) |
| **Comorbidities** | |
| Cardiovascular disease | 25 (50.0) |
| Lung disease other than COPD | 8 (16.0) |
| Dyslipidemia | 19 (38.0) |
| Diabetes mellitus | 6 (12.0) |
| Liver disease | 6 (12.0) |
| Kidney disease | 3 (6.0) |

Data shown as mean ± SD or n (%).

kg = kilogram, m = meter, $FEV_1$ = forced expiratory volume in 1 second, FVC = forced vital capacity, L = liter, LABA = Long-acting beta2-agonist, LAMA = Long-acting muscarinic antagonist, SAMA = Short-acting beta-agonist, SAMA = Short-acting muscarinic-antagonist, ICS = Inhaled corticosteroid, COPD = chronic obstructive pulmonary disease.

Interestingly, our COPD patients had a little above 2 points of Borg dyspnea scale after exercise testing (see Table 2). This finding may result from low dyspnea perception and high symptom variability of COPD patients. There are several established evidences of poor correlation between symptom perception and $FEV_1$ [12]. COPD symptoms show high seasonal, weekly, and daily variability [12]. Moreover, oxygen saturation did not differ before and after exercise testing in our study. It is possible that our COPD patients had less-severe airway obstruction ($FEV_1$ of 63%), therefore desaturation after walking test is not shown according to an established definition; a fall in $SpO_2 \geq 4\%$ or $SpO_2 < 90\%$ [13]. 6MWDs in our patients (around 350 m) are similar to another study on stable COPD patients in Thailand [14]. In this study, the majority of COPD patients were spirometric grade 2 (mean $FEV_1$ of 66%) with a mean 6MWD of 317 m [14].

**Table 2. The effects of walking length on total distance and physiologic variables.**

| Variables | 20 meters | 30 meters | P-value |
|---|---|---|---|
| 6MWD, meters | 337.82±71.80 | 359.85±77.25 | <0.001 |
| Number of turns, times | 16.46±3.51 | 11.58±2.56 | <0.001 |
| **Before 6MWT** | | | |
| Systolic blood pressure, mmHg | 130.26±18.04 | 128.56±16.78 | 0.220 |
| Diastolic blood pressure, mmHg | 79.56±11.19 | 80.52±12.83 | 0.196 |
| Borg dyspnea scale | 0.75±0.71 | 0.81±0.65 | 0.060 |
| SpO$_2$, % | 97.17±1.50 | 97.10±1.36 | 0.699 |
| Heart rate, bpm | 79.64±11.99 | 80.06±13.07 | 0.150 |
| Respiratory rate, bpm | 20.16±3.89 | 19.80±2.95 | 0.237 |
| **After 6MWT** | | | |
| Systolic blood pressure, mmHg | 139.56±19.39 | 134.94±21.40 | 0.019 |
| Diastolic blood pressure, mmHg | 83.22±12.82 | 81.02±21.40 | 0.210 |
| Borg dyspnea scale | 2.29±0.81 | 2.39±0.99 | 0.280 |
| SpO$_2$, % | 96.14±2.41 | 96.14±2.37 | 1.000 |
| Heart rate, bpm | 86.16±14.17 | 88.24±14.25 | 0.077 |
| Respiratory rate, bpm | 22.68±3.47 | 22.88±3.19 | 0.547 |

Data shown as mean ± SD.

6MWD = six-minute walk distance, mmHg = millimeters of mercury, SpO$_2$ = oxygen saturation using a pulse oximeter, bpm = beats per minute.

Beside shorter walkway had more turns than longer walkway. Most of physical variables after tests were higher than before tests due to increased physical activity in our study. These results indicated that the short walkway length affects physiologic changes and decrease in 6MWD.

**Table 3. The effects of walking length on physiologic variables after 6MWT.**

| Variables | Before 6MWT | After 6MWT | P-value |
|---|---|---|---|
| **20-m walkway length** | | | |
| Systolic blood pressure, mmHg | 130.26±18.04 | 139.56±19.39 | <0.001 |
| Diastolic blood pressure, mmHg | 79.56±11.19 | 83.22±12.82 | 0.003 |
| Borg dyspnea scale | 0.65±0.71 | 2.29±0.81 | <0.001 |
| SpO$_2$, % | 97.14±1.50 | 96.14±2.41 | <0.001 |
| Heart rate, bpm | 77.64±11.99 | 86.16±14.17 | <0.001 |
| Respiratory rate, bpm | 20.16±3.79 | 22.68±3.47 | <0.001 |
| **30-m walkway length** | | | |
| Systolic blood pressure, mmHg | 125.56±16.78 | 134.94±21.40 | <0.001 |
| Diastolic blood pressure, mmHg | 80.52±12.83 | 81.02±15.02 | 0.816 |
| Borg dyspnea scale | 0.81±0.65 | 2.39±0.99 | <0.001 |
| SpO$_2$, % | 97.10±13.6 | 96.14±2.37 | <0.001 |
| Heart rate, bpm | 80.06±13.07 | 88.24±14.25 | <0.001 |
| Respiratory rate, bpm | 19.80±2.95 | 22.88±3.19 | <0.001 |

Data shown as mean ± SD.

6MWT = six-minute walk test, mmHg = millimeters of mercury, SpO$_2$ = oxygen saturation using a pulse oximeter, bpm = beats per minute.

Our study had a few limitations. To decrease the learning effect in walking tests, participants were randomized to short or long walkway length in the first test, then switched to the other test. Moreover, to assure accurate test results, participants had to rest between tests until all physiologic and dyspnea variables had stabilized. However, 6MWD may be expected to increase as participants become increasingly familiar with the exercise.

## Conclusion

The walkway length had significant effect on walking distance in COPD patients. A 30-m walkway length should still be recommended in 6MWT for COPD assessment.

## Supporting information

**S1 Checklist.**
(DOC)

**S1 File. Study protocol in English.**
(PDF)

**S2 File. Study protocol in Thai.**
(PDF)

**S3 File. Information sheet in English.**
(PDF)

**S4 File. Information sheet in Thai.**
(PDF)

**S5 File. Consent form in English.**
(PDF)

**S6 File. Consent form in Thai.**
(PDF)

**S7 File. Dataset of 6-minute walk test in COPD patients.**
(XLSX)

## Acknowledgments

The authors would like to thank Michael Jan Everts and Dr Kanon Jatuworapruk, Faculty of Medicine, Thammasat University, for proofreading this manuscript.

## Author Contributions

**Conceptualization:** Narongkorn Saiphoklang, Apiwat Pugongchai.

**Data curation:** Apiwat Pugongchai, Kanyada Leelasittikul.

**Formal analysis:** Narongkorn Saiphoklang, Apiwat Pugongchai.

**Funding acquisition:** Apiwat Pugongchai.

**Investigation:** Narongkorn Saiphoklang, Kanyada Leelasittikul.

**Methodology:** Narongkorn Saiphoklang.

**Project administration:** Narongkorn Saiphoklang, Apiwat Pugongchai.

**Resources:** Narongkorn Saiphoklang, Apiwat Pugongchai, Kanyada Leelasittikul.

**Software:** Apiwat Pugongchai.

**Supervision:** Narongkorn Saiphoklang.

**Validation:** Narongkorn Saiphoklang, Apiwat Pugongchai, Kanyada Leelasittikul.

**Visualization:** Narongkorn Saiphoklang.

**Writing – original draft:** Apiwat Pugongchai, Kanyada Leelasittikul.

**Writing – review & editing:** Narongkorn Saiphoklang.

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
