## [Decision Letter · Decision Letter 0]

19 Oct 2021

PONE-D-21-27964Comparison between 20 and 30 meters in walkway length affecting the 6-minute walk test in patients with chronic obstructive pulmonary disease: a randomized crossover studyPLOS ONE

Dear Dr. Saiphoklang,

Thank you for submitting your manuscript to PLOS ONE. After careful consideration, we feel that it has merit but does not fully meet PLOS ONE’s publication criteria as it currently stands. Therefore, we invite you to submit a revised version of the manuscript that addresses the points raised during the review process.

We look forward to receiving your revised manuscript.

Kind regards,

Davor Plavec, MD, MSc, PhD, Prof.

Academic Editor

PLOS ONE

Journal Requirements:

2. We note that you have selected “Clinical Trial” as your article type. PLOS ONE requires that all clinical trials are registered in an appropriate registry (the WHO list of approved registries is at      https://www.who.int/clinical-trials-registry-platform/network/primary-registries" https://www.who.int/clinical-trials-registry-platform/network/primary-registries" https://www.who.int/clinical-trials-registry-platform/network/primary-registries and more information on trial registration is at http://www.icmje.org/about-icmje/faqs/clinical-trials-registration/).

Please state the name of the registry and the registration number (e.g. ISRCTN or ClinicalTrials.gov) in the submission data and on the title page of your manuscript.

a) Please provide the complete date range for participant recruitment and follow-up in the methods section of your manuscript.

b) If you have not yet registered your trial in an appropriate registry, we now require you to do so and will need confirmation of the trial registry number before we can pass your paper to the next stage of review. Please include in the Methods section of your paper your reasons for not registering this study before enrolment of participants started. Please confirm that all related trials are registered by stating: “The authors confirm that all ongoing and related trials for this drug/intervention are registered”.

Please see http://journals.plos.org/plosone/s/submission-guidelines#loc-clinical-trials for our policies on clinical trials.

“The work was supported by Thammasat University Hospital, Thailand. The funders had no role in study design, data collection and analysis, decision to publish, or preparation of the manuscript.”

We note that you have provided funding information within the Acknowledgements Section. Please note that funding information should not appear in the Acknowledgments section or other areas of your manuscript. We will only publish funding information present in the Funding Statement section of the online submission form.

“AP received the research fund of Thammasat University Hospital, Thailand.

Reviewers' comments:

Reviewer's Responses to Questions

**Comments to the Author**

1. Is the manuscript technically sound, and do the data support the conclusions?

Reviewer #1: Yes

Reviewer #2: Yes

Reviewer #3: Yes

2. Has the statistical analysis been performed appropriately and rigorously? 

Reviewer #1: Yes

Reviewer #2: I Don't Know

Reviewer #3: Yes

3. Have the authors made all data underlying the findings in their manuscript fully available?

Reviewer #1: Yes

Reviewer #2: Yes

Reviewer #3: Yes

4. Is the manuscript presented in an intelligible fashion and written in standard English?

Reviewer #1: Yes

Reviewer #2: Yes

Reviewer #3: Yes

5. Review Comments to the Author

Reviewer #1: Few comments:

Introduction

- the parameter FEV1/FVC is not usual to express in percentage, rather is expressed in decimal number: not < 70%, but 0.70

- METHODS. It is not written which Borg scale was used, with 10 or 11 points – it should be added.

- It is not emphasized wheather patient was encouraged or not, because it is important in a methodology of 6MWT.

- RESULTS

- In Table No 1, I suggest to delete in Legend PDE4=phosphodiesterase 4, because there was none patient with such therapy

- It is very unusual that, as we can see in table 2, the Borg Dyspnea scale was just a little above 2 points after exercize, especially because the walking distance was rather short, around 350 m, when most of COPD patients has more pronounced dyspnea...It is also unexpected that oxygen saturation did not differ befere and after exercize while such short distance was achieved. Also when FEV1 in COPD is arround 63%, we could expect longer distance in 6MWT. Please, comment that in a discussion.

Reviewer #2: The article can benefit from some minor grammatical and syntax corrections.

Example: “ However, there are few published data on 6MWT in the COPD

patient population“ should be rephrased because there are 718 results in PubMed for 6MWT and COPD.

The population in study (ethnicity) should be stated.

Reviewer #3: This is a well-designed crossover study. Randomization and sample size are clearly documented. But there is a significant problem in the introduction: the sentence is "We hypothesized that the results would not

differ between the two distances." In that case, it is an equivalence study, which must be powered and analyzed completely differently. So what is it? Clearly, the sample size computations indicate a difference. Please clarify.

1. Were the parameters estimated in the sample size realized in the trial? Please add to the discussion.

2. p<.05 has little meaning in the presence of so many hypothesis tests, as does the term "significance". Please modify the description to indicate that type I error rate is preserved for the primary outcome, and all other tests are exploratory in nature.

6. PLOS authors have the option to publish the peer review history of their article (what does this mean?). If published, this will include your full peer review and any attached files.

Reviewer #1: **Yes: **Prof. Sanja Popović-Grle, M.D., Ph.D.

Reviewer #2: No

Reviewer #3: No

---

## [Author Response · Author response to Decision Letter 0]

5 Nov 2021

Dear Reviewer of PLOS ONE:

I submitted an original contribution to PLOS ONE entitled “Comparison between 20 and 30 meters in walkway length affecting the 6-minute walk test in patients with chronic obstructive pulmonary disease: a randomized crossover study” (Manuscript number: PONE-D-21-27964).

Thank you so much for your great reviews and comments regarding my manuscript. I would like to explain and submit a revised manuscript. My response to the comment is below.

RESPONSE TO REVIEWER 1:

REVIEWER #1 COMMENT 1: Introduction - the parameter FEV1/FVC is not usual to express in percentage, rather is expressed in decimal number: not < 70%, but 0.70

RESPONSE: I have corrected this number in the Method section, Study design and participants. Thank you for kindly reviews.

REVIEWER #1 COMMENT 2: METHODS. It is not written which Borg scale was used, with 10 or 11 points – it should be added.

RESPONSE: I have added these words in the Method section, Procedures and outcomes. 

REVIEWER #1 COMMENT 3: It is not emphasized whether patient was encouraged or not, because it is important in a methodology of 6MWT.

RESPONSE: I have added more information about encouragement in the Method section, Procedures and outcomes. 

REVIEWER #1 COMMENT 4: RESULTS - In Table No 1, I suggest to delete in Legend PDE4=phosphodiesterase 4, because there was none patient with such therapy

RESPONSE: I have removed this word from footnote in Table 1. 

REVIEWER #1 COMMENT 5: It is very unusual that, as we can see in table 2, the Borg Dyspnea scale was just a little above 2 points after exercise, especially because the walking distance was rather short, around 350 m, when most of COPD patients has more pronounced dyspnea...It is also unexpected that oxygen saturation did not differ before and after exercise while such short distance was achieved. Also when FEV1 in COPD is around 63%, we could expect longer distance in 6MWT. Please, comment that in a discussion.

RESPONSE: I have added more discussion in the Discussion section, Paragraph 3.

RESPONSE TO REVIEWER 2:

REVIEWER #2 COMMENT 1: The article can benefit from some minor grammatical and syntax corrections.

Example: “However, there are few published data on 6MWT in the COPD patient population” should be rephrased because there are 718 results in PubMed for 6MWT and COPD.

RESPONSE: I have reorganized the sentence in the Introduction section, Paragraph 2. Thank you so much for kindly reviews.

REVIEWER #2 COMMENT 2: The population in study (ethnicity) should be stated.

RESPONSE: I have added this word in the Introduction section, Paragraph 2. 

RESPONSE TO REVIEWER 3:

REVIEWER #3 COMMENT 1: This is a well-designed crossover study. Randomization and sample size are clearly documented. But there is a significant problem in the introduction: the sentence is "We hypothesized that the results would not differ between the two distances." In that case, it is an equivalence study, which must be powered and analyzed completely differently. So what is it? Clearly, the sample size computations indicate a difference. Please clarify.

1. Were the parameters estimated in the sample size realized in the trial? Please add to the discussion.

RESPONSE: I have added more discussion about power of the test in the Discussion section, Paragraph 1. Thank you so much for wonderful reviews.

REVIEWER #3 COMMENT 2: 

2. p<.05 has little meaning in the presence of so many hypothesis tests, as does the term "significance". Please modify the description to indicate that type I error rate is preserved for the primary outcome, and all other tests are exploratory in nature.

RESPONSE: I have corrected the description of secondary outcomes in the Results section, Paragraph 3. Thank you for kindly suggestions.

I appreciate your consideration of my manuscript for publication in PLOS ONE.

Sincerely,

Narongkorn Saiphoklang, M.D.

---

## [Decision Letter · Decision Letter 1]

19 Nov 2021

PONE-D-21-27964R1Comparison between 20 and 30 meters in walkway length affecting the 6-minute walk test in patients with chronic obstructive pulmonary disease: a randomized crossover studyPLOS ONE

Dear Dr. Saiphoklang,

Thank you for submitting your manuscript to PLOS ONE. After careful consideration, we feel that it has merit but does not fully meet PLOS ONE’s publication criteria as it currently stands. Therefore, we invite you to submit a revised version of the manuscript that addresses the points raised during the review process.

Please revise as suggested by reviewers.==============================

We look forward to receiving your revised manuscript.

Kind regards,

Davor Plavec, MD, MSc, PhD, Prof.

Academic Editor

PLOS ONE

Journal Requirements:

Additional Editor Comments (if provided):

Please revise as suggested by the reviewers.

Reviewers' comments:

Reviewer's Responses to Questions

**Comments to the Author**

1. If the authors have adequately addressed your comments raised in a previous round of review and you feel that this manuscript is now acceptable for publication, you may indicate that here to bypass the “Comments to the Author” section, enter your conflict of interest statement in the “Confidential to Editor” section, and submit your "Accept" recommendation.

Reviewer #1: All comments have been addressed

Reviewer #2: (No Response)

Reviewer #3: (No Response)

2. Is the manuscript technically sound, and do the data support the conclusions?

Reviewer #1: Yes

Reviewer #2: Yes

Reviewer #3: (No Response)

3. Has the statistical analysis been performed appropriately and rigorously? 

Reviewer #1: Yes

Reviewer #2: I Don't Know

Reviewer #3: (No Response)

4. Have the authors made all data underlying the findings in their manuscript fully available?

Reviewer #1: Yes

Reviewer #2: Yes

Reviewer #3: (No Response)

5. Is the manuscript presented in an intelligible fashion and written in standard English?

Reviewer #1: Yes

Reviewer #2: No

Reviewer #3: (No Response)

6. Review Comments to the Author

Reviewer #1: All questions and comments by me were included as good answers and reasonalble explantations, also methods are improved, so I find the manuscipt ready for publishing.

Reviewer #2: When I suggest the linguistic and semantic corrections i mean that the whole article should be rechecked for consistence. I can still find sentences that are at least not precise like: They found that 6MWD with 30-m walkway had significantly longer than 20-m walkway in COPD patients (mean difference in 22.1 m)

That should sound like: "They found that, in COPD patients, 6MWD with 30-m walkway is significantly longer than with 20-m walkway (mean difference of 22.1 m) [10]

Please recheck the whole article and correct such sentences. I have no other comments and the article is worth publishing.

Regards

Reviewer #3: The authors seem not to understand my comment on the last part of the Introduction where they say that we hypothesized that there would be NO difference. Clearly you found a difference, you powered for a difference, yet the sentence is still there. Either remove the sentence or change it to reflect that you are hypothesizing that there WILL BE a difference.

7. PLOS authors have the option to publish the peer review history of their article (what does this mean?). If published, this will include your full peer review and any attached files.

Reviewer #1: **Yes: **prof. dr.sc. Sanja Popović-Grle, M.D.Ph.D.

Reviewer #2: No

Reviewer #3: No

---

## [Author Response · Author response to Decision Letter 1]

23 Nov 2021

RESPONSE TO REVIEWER:

REVIEWER #1: All questions and comments by me were included as good answers and reasonable explanations, also methods are improved, so I find the manuscript ready for publishing. 

RESPONSE: Thank you so much for wonderful reviews and accepting for my corrections.

REVIEWER #2: When I suggest the linguistic and semantic corrections. I mean that the whole article should be rechecked for consistence. I can still find sentences that are at least not precise like: They found that 6MWD with 30-m walkway had significantly longer than 20-m walkway in COPD patients (mean difference in 22.1 m)

That should sound like: "They found that, in COPD patients, 6MWD with 30-m walkway is significantly longer than with 20-m walkway (mean difference of 22.1 m) [10]

Please recheck the whole article and correct such sentences. I have no other comments and the article is worth publishing.

RESPONSE: I have corrected this sentence in the Discussion section. Thank you for kindly suggestions.

REVIEWER #3: The authors seem not to understand my comment on the last part of the Introduction where they say that we hypothesized that there would be NO difference. Clearly you found a difference, you powered for a difference, yet the sentence is still there. Either remove the sentence or change it to reflect that you are hypothesizing that there WILL BE a difference.

RESPONSE: I have removed the sentence to reflect the hypothesis in the Introduction section. Thank you for great suggestions and wonderful reviews.

I appreciate your consideration of my manuscript for publication in PLOS ONE.

Sincerely,

Narongkorn Saiphoklang, M.D.

---

## [Decision Letter · Decision Letter 2]

20 Dec 2021

Comparison between 20 and 30 meters in walkway length affecting the 6-minute walk test in patients with chronic obstructive pulmonary disease: a randomized crossover study

PONE-D-21-27964R2

Dear Dr. Saiphoklang,

We’re pleased to inform you that your manuscript has been judged scientifically suitable for publication and will be formally accepted for publication once it meets all outstanding technical requirements.

Kind regards,

Davor Plavec, MD, MSc, PhD, Prof.

Academic Editor

PLOS ONE

Additional Editor Comments (optional):

I am happy to announce that your paper is ready for acceptance with the minor revision suggested by the reviewer #2.

Reviewers' comments:

Reviewer's Responses to Questions

**Comments to the Author**

1. If the authors have adequately addressed your comments raised in a previous round of review and you feel that this manuscript is now acceptable for publication, you may indicate that here to bypass the “Comments to the Author” section, enter your conflict of interest statement in the “Confidential to Editor” section, and submit your "Accept" recommendation.

Reviewer #2: All comments have been addressed

Reviewer #3: All comments have been addressed

2. Is the manuscript technically sound, and do the data support the conclusions?

Reviewer #2: Yes

Reviewer #3: (No Response)

3. Has the statistical analysis been performed appropriately and rigorously? 

Reviewer #2: I Don't Know

Reviewer #3: (No Response)

4. Have the authors made all data underlying the findings in their manuscript fully available?

Reviewer #2: Yes

Reviewer #3: (No Response)

5. Is the manuscript presented in an intelligible fashion and written in standard English?

Reviewer #2: Yes

Reviewer #3: (No Response)

6. Review Comments to the Author

Reviewer #2: Still minor problem. Please correct: "It is possible that our COPD patients were nonsevere airway obstruction (FEV1 of 63%)," to: "It is possible that our COPD patients had less-severe airway obstruction (FEV1 of 63%)" or something like that and there are no further objections from my point.

Regards

Reviewer #3: (No Response)

7. PLOS authors have the option to publish the peer review history of their article (what does this mean?). If published, this will include your full peer review and any attached files.

Reviewer #2: No

Reviewer #3: No

---

## [Editor Report · Acceptance letter]

30 Dec 2021

PONE-D-21-27964R2 

Comparison between 20 and 30 meters in walkway length affecting the 6-minute walk test in patients with chronic obstructive pulmonary disease: a randomized crossover study 

Dear Dr. Saiphoklang:

I'm pleased to inform you that your manuscript has been deemed suitable for publication in PLOS ONE. Congratulations! Your manuscript is now with our production department. 

Kind regards, 

on behalf of

Dr. Davor Plavec 

Academic Editor

PLOS ONE